# Research on the impact of enterprise mergers and acquisitions on technological innovation: An empirical analysis based on listed Chinese enterprises

Yujiao Bai[1¤a], Hao Zhang[2¤b]*

1 Center for Industrial and Business Organization, Dongbei University of Finance and Economics, Dalian, Liaoning, China, 2 School of Economics and Management, Huaibei Normal University, Huaibei, Anhui, China

¤a Current address: Dongbei University of Finance and Economics, Dalian City, Liaoning Province, China
¤b Current address: Huaibei Normal University, Huaibei City, Anhui Province, China
* zhangh@chnu.edu.cn

**Data Availability Statement:** All relevant data are within the manuscript and its Supporting Information files.

## Abstract

As an important means for enterprises to acquire technological resources, the impact of mergers and acquisitions on technological innovation and underlying mechanisms deserve in-depth study. Using the merger and acquisition data of A-share listed Chinese companies from 2007 to 2020 in Shanghai and Shenzhen, the causal effects and influence mechanisms between mergers and acquisitions and technological innovation are identified and tested using the Difference-in-Differences method. The study finds that mergers and acquisitions have a long-term, sustained, technological innovation-enhancing effect on firms. Mechanism tests show that mergers and acquisitions can promote the technological innovation of enterprises by improving production efficiency, enriching digital knowledge, and enhancing market power. A heterogeneity analysis shows that the effect of mergers and acquisitions in enhancing technological innovation is more significant when the mergers and acquisitions meet domestic merger and acquisition requirements, when there is a small transaction size, and when the enterprises involved in the mergers and acquisitions are not state-owned. It is suggested that enterprises and the government should use multiple measures, while considering the impact of heterogeneity, to take full advantage of the positive effects of mergers and acquisitions on technological innovation.

## 1. Introduction

In 2010, China became the country with the greatest involvement in the manufacturing industry; however, it has long faced the difficulties of serious duplication of industry construction, low industrial concentration, a weak independent innovation ability, and weak market competitiveness. Under the pattern of latecomer competition, breaking through the technological blockade, solving the "bottleneck" problem, and achieving subversive innovation are necessary strategies for Chinese enterprises to catch up with current technology. However, due to the

**Funding:** This research was supported by Anhui Province Excellent Youth Research Project in Universities, grant number 2023AH030082. The funders had no role in study design, data collection and analysis, decision to publish, or preparation of the manuscript.

**Competing interests:** The authors have declared that no competing interests exist.

lack of technical skills and the long technology research and development cycle, it is a relatively slow method for enterprises to rely on independent innovation to improve their technological levels [1, 2]. According to the theory of open innovation, enterprises can rapidly acquire heterogeneous knowledge that is difficult to purchase in the factor market through mergers and acquisitions [3], and external digital resources such as talent, technology and knowledge acquired through mergers and acquisitions can be utilized to improve the level of technological innovation of enterprises [4]. With the growing demand for technological innovation in China, enterprises need to absorb new technological knowledge through cross-industry integration to more quickly achieve technological innovation in order to maintain competitive advantages and make use of new opportunities [5]; thus, a large number of enterprises have started to acquire external knowledge through mergers and acquisitions to enhance technological innovation abilities.

According to CSMAR data, in terms of quantity and amount, China's merger and acquisition activities have increased over the past decade [6]. In 2020, the number of China's merger and acquisition transactions reached a record high, with a total of 9,879 merger and acquisition events and a total merger and acquisition value of $409.969 billion, up 168% and 356%, respectively, compared with 2007. However, there are many cases of failure. For example, since 2015, Chengdu Tianxiang Environment Co., Ltd., China's leading environmental protection equipment company, has acquired the U.S. sludge treatment equipment company Centrisys Corporation, Germany's Belfinger Water Treatment (BWT), and Germany's "solid waste giant" Euromonitor Group, but the company went bankrupt and delisted in December 2020 due to a lack of breakthroughs in core technology and poor operations.

Based on the above, this study aims to answer the following questions: Can mergers and acquisitions promote the technological innovation of Chinese enterprises? If so, what is the mechanism? As there are both state-owned and non-state-owned enterprises in China, does the relationship between mergers and acquisitions and technological innovation reflect heterogeneity according to the characteristics of enterprises and types of mergers and acquisitions? To address these questions, this paper identifies and examines the causal effect and influence mechanism between mergers and acquisitions and technological innovation based on the merger and acquisition data of Chinese A-share listed firms in Shanghai and Shenzhen under the framework of the Difference-in-Differences method of empirical testing. This paper further examines the impact of heterogeneity, i.e., firm idiosyncrasies, according to the type of merger and acquisition transactions and the nature of firm ownership.

## 2. Literature review

### 2.1 Mergers and acquisitions

As a business strategy for enterprises to expand scale, increase market share, optimize resources, and reduce costs, the merger and acquisition effect is a core issue of concern for many parties. The literature on the effects of mergers and acquisitions mainly focuses on merger and acquisition firms and explores the impact of mergers and acquisitions on their market power, productivity and wealth [7–10]. Tang et al. (2022) [11] found that mergers and acquisitions can significantly enhance the market power of manufacturing firms from the perspective of digital transformation. Jiang (2021) [12] utilized Thomson Reuters mergers and acquisitions data from 2003 to 2007 and merged it with a database of Chinese industrial enterprises and found that mergers and acquisitions in China not only enhanced the market power of the enterprises but also significantly improved their productivity. Focusing on the Chinese coal industry, He et al (2020) [13] found that mergers and acquisitions by listed coal companies lead to efficiency gains. Using data from the U.S. manufacturing industry, Kim et al.

(2019) [14] found that the mergers and acquisition experience of merger and acquisition firms is most positively correlated with productivity. Fich et al. (2018) [10] found that shareholders of merger and acquisition firms gained wealth in merger and acquisition transactions.

## 2.2 Technological innovation

Technological innovation is a key driving force for enterprises to gain and maintain competitiveness and performance [15]. This is also a necessary condition for promoting sustainable economic growth in countries, especially emerging countries such as China. However, technological innovation is characterized by long investment returns, high costs, strong challenges, and high uncertainty [16]. Therefore, studying how enterprises can achieve technological innovation has attracted widespread attention from all sectors. Scholars have conducted extensive discussions of the factors that affect corporate technological innovation, mainly focusing on knowledge management [17, 18], corporate governance [19], and tax policies [20]. For example, Li et al. (2023) [19] used panel data from 30 provinces in China from 2013 to 2020 and found that promoting digitalization in manufacturing can promote technological innovation. From the perspective of related industry layouts, Qian et al. (2023) [21] verified that the agglomeration of the technology service industry can promote enterprise technological innovation. Kanojia et al. (2023) [22] used data from 5747 Indian companies to explore the impact of external and internal factors on technological innovation.

## 2.3 Mergers and acquisitions and technological innovation

In recent years, the relationship between mergers and acquisitions and technological innovation has gradually received attention from academics, and some studies have shown that mergers and acquisitions can drive technological innovation, and merger and acquisition firms gain access to the technological resources of the acquired firms through acquisitions to drive the technological change of the merger and acquisition firms after acquisitions [23–26]. Wang et al. (2022) [27] found that firms carry out the redistribution of innovations through mergers and acquisitions and increase their level of innovation. Bena et al. (2014) [28] argued that innovation often leads to merger and acquisition synergies and that synergies from merger and acquisition activities promote innovation output [29, 30]. However, some studies have shown that mergers and acquisitions lead to reduced innovation efficiency [31] and inhibit firms' innovation [32]. Haucap et al. (2019) [33], using data from the European pharmaceutical industry, showed that mergers and acquisitions inhibit firms' R&D investment and reduce the level of firms' technological innovation. Using data from China's manufacturing industry as a sample, Ma et al. (2017) [34] empirically found that nontechnology mergers and acquisitions have a negative impact on the technological innovation of merger and acquisition firms. The empirical results of the impact of mergers and acquisitions on technological innovation are inconsistent, with most studies suggesting that they are influenced by factors such as the motives for mergers and acquisitions and firm and industry heterogeneity. In general, in nontechnology mergers and acquisitions, the acquired firms tend not to provide technical support to the acquiring firms and thus often do not contribute to the technological innovation of the acquiring firms [35].

## 2.4 Literature summary

Scholars have conducted rich research on the economic consequences of mergers and acquisitions and the driving force of corporate technological innovation; however, the relationship between mergers and acquisitions and technological innovation can still be further studied based on the following three points. First, for the empirical analysis of different research

objects, scholars have found that mergers and acquisitions may promote enterprise innovation or may have a negative impact; therefore, the impact of mergers and acquisitions on technological innovation needs to be further tested. Second, the research on the impact and mechanism of mergers and acquisitions on technological innovation is still quite limited, and the path of how to realize technological innovation through mergers and acquisitions is still unclear. Third, there is a lack of research on whether the impact of mergers and acquisitions on technological innovation varies according to firm ownership traits and merger and acquisition type.

In comparison with the literature, the contributions of this study are as follows. First, the impact of mergers and acquisitions on technological innovation is examined using Chinese listed firms as the research object. Second, this study further expands the research on the mechanism and path of mergers and acquisitions on technological innovation and improves the theoretical framework of mergers and acquisitions for technological innovation. Third, this study captures the heterogeneous results of mergers and acquisitions for technological innovation by grouping regressions based on the relevant factors affecting the promotion effect, such as the type of merger and acquisition, the intensity of the merger and acquisition, and the nature of firm ownership.

## 3. Theoretical analysis and research hypotheses

### 3.1 The impact of corporate mergers and acquisitions on technological innovation

An innovative model is based on the research of Chen et al. (2019) [36]. Assuming a market with two companies ($i,l$), the inverse demand function for both companies is $P = a−Q$, where $Q$ is the market's product output. Assuming that the fixed cost of the enterprise is zero, it is assumed that the enterprise has always made innovation investments ($x_i$, $x_j$) since its establishment; that is, $x_i>0, x_j>0$, $x_i$, $x_j$ follows the law of diminishing marginal returns and, thus, appears in quadratic form. Next, we compare and analyze the changes in innovation before and after corporate mergers and acquisitions.

**3.1.1 No mergers or acquisitions.** When there is no merger or acquisition, the cost of company $i$ is:

$$C_i^T = (C - x_i)q_i + \frac{1}{2}\gamma_i x_i^2 \tag{1}$$

Here, $\gamma_i$ represents the cost efficiency of enterprise $i$'s technological innovation, and $\gamma_i$ decreases with an increase in enterprise $i$'s technological innovation level. The profit of company $i$ is:

$$\pi_i = (a - Q)q_i - \left[(C - x_i)q_i + \frac{1}{2}\gamma_i x_i^2\right] \tag{2}$$

By taking the first derivative of $q_i$ and $x_i$ in Formula (2) in sequence, the equilibrium value of $x_i$ can be obtained:

$$x_i^* = \frac{4(a - C)}{9\gamma_i - 4} \tag{3}$$

The smaller $\gamma_i$ is, the larger $x_i^*$ is, which means that a higher level of enterprise technological innovation is associated with enterprises making more investments in technological innovation. However, when enterprises' technological innovation reaches a certain level, that is, $0<\gamma_i<4/9$, the enterprise no longer invests in technological innovation.

**3.1.2 Mergers and acquisitions.** When company $i$ acquires company $l$, the cost after the acquisition of company $i$ is:

$$C_i^T = \left[ C - (x_i + \varphi_l x_l) q_i \right] + \frac{1}{2} \gamma_i (x_i^2 + \delta x_l^2) \tag{4}$$

Here, $\varphi_l$ represents the absorption and digestion level of innovation resources by company $i$ to company $l$, which depends on the resource integration and digestion ability of enterprise $i$ after a merger or acquisition and is consistent with $\varphi_l > 0$; $\delta$ represents the cost of enterprise $i$ absorbing technology and knowledge from enterprise $l$, which is consistent with $\delta > 0$. The profit of enterprise $i$ is:

$$\pi_i = (a - Q - C + x_i + \varphi_l x_l) q_i - \frac{1}{2} \gamma_i (x_i^2 + \delta x_l^2) \tag{5}$$

By taking the first derivative of $q_i$ and $x_i$ in Formula (5) in sequence, the equilibrium value of $x_i$ can be obtained:

$$x_i^* = \frac{a - C + \varphi_l x_l}{2\gamma_i - 1} \tag{6}$$

According to Formula (6), if an enterprise still invests in technological innovation after a merger or acquisition, it needs to meet $\gamma_i > 1/2$; when $\gamma_i < 1/2$, that is, $x_i^* < 0$, mergers and acquisitions inhibit the technological innovation of enterprise $i$.

On the basis of the previous text, we compare the changes in $x_i$ before and after the merger:

$$\triangle x_i^* = \frac{(a - C)\gamma_i + (9\gamma_i - 4)\varphi_l x_l}{(2\gamma_i - 1)(9\gamma_i - 4)} \tag{7}$$

When the enterprise's technological innovation is low, to absorb the new technologies obtained from mergers and acquisitions, the enterprise must continue to increase its R&D efforts, enhance its ability to absorb new technologies, and improve its technological innovation level. When a company has high technological innovation, it acquires technological knowledge through acquisitions. An acquiring party with a higher level of technological innovation can innovate these new technologies, which can further enhance the company's degree of technological innovation. When the technological innovation in enterprises is moderate, mergers and acquisitions often suppress the driving force of technological innovation under the dual pressure of having to digest and absorb new technologies and continuous independent innovation. Based on this, the following research hypothesis is proposed.

Hypothesis 1: Because the average technological innovation capability of Chinese enterprises is relatively low, mergers and acquisitions can promote technological innovation.

## 3.2 Analysis of impact mechanisms

**3.2.1 Path to improving production efficiency.** Mergers and acquisitions can improve enterprises' production efficiency, thereby enhancing their technological innovation. Through mergers and acquisitions, enterprises can achieve optimized resource allocation and improve production efficiency [37]. Similarly, Chen et al. (2019) [36] found that Chinese manufacturing companies can improve their production efficiency and promote innovation through mergers and acquisitions. After the merger and acquisition of enterprises has been completed, the integration of the enterprise's original technology with the digital technology and resources of the acquired enterprise will facilitate more intelligent and efficient use of digital technology

for production [38], improve the overall efficiency of the enterprise's production, increase the enterprise's R&D investment, and enhance the enterprise's level of technological innovation. In summary, the following research hypothesis is proposed:

Hypothesis 2: Mergers and acquisitions can promote technological innovation by improving corporate productivity.

**3.2.2 Path to enriching digital knowledge.** Corporate mergers and acquisitions empower digital transformation, increase intangible assets, and enhance knowledge spillover effects; thus, they have a positive impact on technological innovation in enterprises. Corporate mergers and acquisitions can facilitate the direct acquisition of digital assets and external knowledge, allow for quick compensation for shortcomings in digital resources, help enterprises achieve digital transformation [39], increase the stock of enterprise knowledge, improve enterprises' ability to receive new external knowledge, and accelerate the speed of integration of the knowledge resources of acquiring enterprises [40, 41]. With such mergers and acquisitions, enterprises can more easily leverage newly formed digital capabilities to expand their potential knowledge portfolio, thereby enhancing their inclination toward technological innovation [42, 43]. Therefore, the increase in basic knowledge generated by corporate mergers and acquisitions helps improve enterprises' ability to integrate innovative resources, thereby promoting technological innovation. In summary, the following research hypothesis is proposed:

Hypothesis 3: Mergers and acquisitions can promote technological innovation by enriching a company's digital knowledge.

**3.2.3 Path to enhancing market power.** Mergers and acquisitions can enhance a company's market power, thereby enhancing its degree of technological innovation. First, through mergers and acquisitions, enterprises can share resources such as human capital, intangible assets, and advanced technology, which to a certain extent generates synergy in management and technological R&D [44]. Moreover, the increase in market power generated by mergers and acquisitions can accelerate the speed of enterprise technological integration, strengthen enterprise R&D efforts [6], and increase the degree of enterprise innovation output. Second, companies eliminate the threat of potential competitors through mergers and acquisitions [45, 46]. To absorb new technologies from competitors and maintain their market position, companies increase their R&D funds, increase their R&D efforts, attach importance to R&D output, and improve their technological innovation. In summary, the following research hypothesis is proposed:

Hypothesis 4: Mergers and acquisitions can promote technological innovation by enhancing a company's market power.

Overall, corporate mergers and acquisitions can directly promote technological innovation. In this process, the production efficiency of the enterprise is improved, technical knowledge is enriched, and the market power of the enterprise is increased. These changes help enterprises improve the quality and efficiency of their R&D processes and enhance their ability to integrate and innovate cross-disciplinary technologies, giving them greater advantages in technological innovation.

## 4. Research design

### 4.1 Data sources

In this article, we summarize Chinese merger and acquisition cases and show that the information on merger and acquisition cases was relatively complete and that the number of cases

increased significantly in 2007. Therefore, Chinese A-share listed companies in Shanghai and Shenzhen from 2007 to 2020 are selected as the research objects. To avoid the impact of outliers on the regression results, all continuous variables are subjected to a 1% truncation process.

The screening of merger and acquisition data is as follows. First, all merger and acquisition transaction events of A-share listed companies in Shanghai and Shenzhen are obtained from the CSMAR database. Referring to the methods for processing merger and acquisition data used by Ren et al. (2017) [6], Liu et al. (2018) [8] and Wang et al. (2023) [47], the sample of financial industry companies is excluded. Second, merger and acquisition samples with restructuring types such as share repurchases, debt restructuring, asset divestments, and production replacements are excluded. Then, merger and acquisition samples with transaction amounts less than one million RMB are excluded. Finally, incomplete information samples are excluded. We obtain 2273 M&A companies, totaling 4631 samples.

The 4631 merger and acquisition transaction events are screened twice to eliminate the effect of noise. According to the merger and acquisition information, based on the date of the first merger and acquisition announcement, the acquiring firms have multiple merger and acquisition events in the same year, one merger and acquisition event is held if the target firms are the same firms, all of the merger and acquisition events are held if the target firms are different, and, finally, a sample of 1,940 M&A firms is obtained after processing.

## 4.2 Empirical model construction

Drawing on the practices of Li (2013) [48] and Li et al. (2016) [49], we establish Eq (8) to verify whether a company's technological innovation capability has improved after mergers and acquisitions.

$$inno_{it} = \alpha_0 + \alpha_1 Treat_{it} * Post_{it} + \sum_{c=1}^{7} \alpha_c Control_{it} + Year_{FE} + Firm_{FE} + Ind_{FE} + Pro_{FE} + \varepsilon_{it} \quad (8)$$

Here, $inno_{it}$ represents the technological innovation level of enterprise $i$ in year $t$. $Treat$ represents whether the enterprise has undergone mergers and acquisitions. For enterprises that have not undergone mergers and acquisitions and for the years before the merger and acquisition, both have a value of 0, whereas for enterprises that have undergone mergers and acquisitions, the value is 1 for the year of the merger and acquisition and subsequent years. If a company underwent multiple mergers and acquisitions in different years, the year of the first merger and acquisition is the processing year. $Control$ represents other control variables; $Year_{FE}$, $Firm_{FE}$, $Ind_{FE}$, and $Pro_{FE}$ represent fixed effects for year, individual, industry, and province, respectively; and $\varepsilon_{it}$ represents random error. The control variables are obtained from the CSMAR database to match the financial data with merger and acquisition samples, and the missing values of the acquirer firms are assigned to 0 after matching (firms where no merger and acquisition has occurred are assigned a value of 0). The specific definitions and descriptions of each variable are shown in Table 1.

## 4.3 Variable description

Dependent variable: Referring to Chen et al. (2019) [36], we examine the changes in technological innovation before and after mergers and acquisitions using the number of patent applications as a measure. In the robustness test section, we refer to the approach of Fang et al. (2022) [5] and use the number of effective patents as the dependent variable for the regression.

Independent variable: Whether or not the business has undergone a merger or acquisition (*Treat*∗*Post*). This variable is obtained by multiplying the virtual variable of enterprise

**Table 1. Meaning and explanation of the variables.**

| Variable type | Variable symbol | Meaning of variables |
|---|---|---|
| Dependent variable | inno | Logarithm of 1 plus the number of patent applications from the mergers and acquisitions company |
| Independent variable | Treat | Whether the mergers and acquisitions company has engaged in a merger (1 for the year of merger and 0 otherwise) |
| | Post | Virtual variable (take the value of 1 for the year of a merger or acquisition and subsequent years) |
| Control variable | ros | Operating profit margin (operating profit/operating revenue) |
| | size | Enterprise size (logarithmic of total assets) |
| | lev | Leverage ratio (total liabilities/total assets) |
| | kint | Capital intensity (total assets/operating income) |
| | bdm | Book to market ratio (total assets/market value) |
| | tobinq | Tobin Q (market value/total assets) |
| | lnwor | Total number of employees (logarithm of 1 plus the number total number of employees) |

grouping ($Treat_{it}$, takes the value of 1 for the year of merger or acquisition and 0 otherwise) by the virtual variable of event impact time ($Post_{it}$, takes the value of 1 for the year of merger and acquisition and subsequent years and 0 otherwise). We take the occurrence of mergers and acquisitions as the core explanatory variable to test whether the innovation performance of the treatment group of enterprises significantly differs from that of the control group of enterprises after mergers and acquisitions and then explore the causal effect of mergers and acquisitions on their technological innovation ability.

Control variables: In accordance with the findings of Bena et al. (2014) [28] and Cheng et al. (2023) [50], we select enterprise-level variables such as operating profit margin (ros), enterprise size (size), leverage ratio (lev), capital intensity (kint), book-to-market ratio (bdm), Tobin Q (tobinq), and total number of employees (lnwor) as control variables to best control for the impact of observable enterprise characteristics on the estimation results. The descriptive statistics of the variables are shown in Table 2.

## 5. Empirical analysis

### 5.1 The impact of corporate mergers and acquisitions on technological innovation

The regression results for the impact of corporate mergers and acquisitions on technological innovation are shown in Table 3. The first and third columns and the second and fourth

**Table 2. Descriptive statistical results of the variables.**

| Variable | (1) | (2) | (3) | (4) | (5) |
|---|---|---|---|---|---|
| | Obs | Mean | Standard Deviation | Minimum | Maximum |
| inno | 68,698 | 0.811 | 1.360 | 0 | 8.830 |
| size | 68,698 | 11.51 | 11.07 | 0 | 26.16 |
| lev | 68,698 | 0.230 | 0.272 | 0 | 0.937 |
| kint | 68,698 | 1.883 | 3.643 | 0 | 27.64 |
| bdm | 68,698 | 0.310 | 0.354 | 0 | 1.097 |
| ros | 68,698 | 0.0672 | 0.169 | -0.848 | 0.582 |
| tobinq | 68,698 | 1.013 | 1.335 | 0 | 6.966 |
| lnwor | 68,698 | 3.946 | 3.905 | 0 | 10.75 |

**Table 3. Impact of corporate mergers and acquisitions on technological innovation.**

| Variable | (1) | (2) | (3) | (4) |
|---|---|---|---|---|
|  | inno | inno | inno | inno |
| Treat*Post | 1.407*** | 0.688*** | 0.536*** | 0.422*** |
|  | (0.0131) | (0.0270) | (0.0117) | (0.0244) |
| Controls | NO | NO | YES | YES |
| Year FE | NO | YES | NO | YES |
| Firm FE | NO | YES | NO | YES |
| Ind FE | NO | YES | NO | YES |
| Pro FE | NO | YES | NO | YES |
| Constant | 0.586*** | 0.701*** | 0.00172 | 0.0625*** |
|  | (0.00524) | (0.00431) | (0.00575) | (0.0157) |
| Observations | 68,698 | 68,698 | 68,698 | 68,698 |
| R−squared | 0.144 | 0.685 | 0.437 | 0.754 |

Notes: t statistics in parentheses

* p < 0.1

** p < 0.05

*** p <0.01. These findings are consistent in the following tables.

columns represent the fixed effects of uncontrolled and controlled years, enterprises, industries, and regions, respectively. The third and fourth columns present the regression results for the controlling variables, such as enterprise operating profit margin (*ros*), enterprise size (*size*), leverage ratio (*lev*), capital intensity (*kint*), and book-to-market ratio (*bdm*). The regression results in columns (1)-(4) of Table 3 show that the coefficients of the core explanatory variables (*Treat*Post*) are significantly positive, indicating that corporate mergers and acquisitions have a positive effect on technological innovation. Therefore, hypothesis 1 proposed in this article holds.

## 5.2 Multiple matching methods for testing the impact of corporate mergers and acquisitions on technological innovation

To further reduce the impact of sample selection bias, we match the treatment group enterprises with 1:1 propensity score matching (PSM). The regression estimation results for the matched samples are shown in column (1) of Table 4. The estimated coefficient of the core explanatory variable *Treat*Post* is significantly positive, once again verifying that mergers and acquisitions have a significant positive impact on corporate innovation. In addition, we replace the matching ratio with 1:3 and rerun the regression. The regression results are shown in column (2), and the core explanatory variable is still significantly positive at the 1% level. We also draw on the approach of Azoulay et al. (2010) [51] and McMullin et al. (2022) [52] and use the coarsened exact matching (CEM) and entropy balance matching (EBM) methods to rematch the samples and further test the core conclusions. The test results reveal that the estimated coefficient of the CEM matching method *Treat*Post* is 0.307, and the coefficient of the EBM matching method *Treat*Post* is 0.292. Both of these values are significantly positive, indicating that the conclusions of this paper are still valid and relatively robust under the various matching methods.

## 5.3 Robustness testing

**5.3.1 Testing of replacement regression methods.** Due to the use of enterprise invention patents as the dependent variable and the assignment of a value of 0 to missing data in this

**Table 4. Multiple matching methods for testing the impact of corporate mergers and acquisitions on technological innovation.**

| Variable | (1) | (2) | (3) | (4) |
|---|---|---|---|---|
| | 1:1PSM | 1:3PSM | CEM | EBM |
| Treat*Post | 0.227*** | 0.410*** | 0.307*** | 0.292*** |
| | (0.0253) | (0.0243) | (0.0406) | (0.0250) |
| Controls | YES | YES | YES | YES |
| Year FE | YES | YES | YES | YES |
| Firm FE | YES | YES | YES | YES |
| Ind FE | YES | YES | YES | YES |
| Pro FE | YES | YES | YES | YES |
| Constant | 0.246*** | 0.0745*** | 0.260*** | 0.214*** |
| | (0.0317) | (0.0161) | (0.0361) | (0.0239) |
| Observations | 32,618 | 66,123 | 14,294 | 68,698 |
| R−squared | 0.750 | 0.756 | 0.805 | 0.743 |

article, a *Tobit* model is used for robustness testing to avoid the impact of missing data on the research conclusions. As shown in Table 5 (1), the regression coefficient of corporate mergers and acquisitions is still significantly positive, and the conclusion obtained is consistent with that in the previous text.

**5.3.2 Substitution test for replacing the dependent variable.** To test the reliability of the estimation results, we use enterprises' number of effective patents as a substitute variable for innovation to test the merger and acquisition effect. The regression results are basically consistent with the benchmark results, and the conclusion is still valid.

**5.3.3 Excluding inspection of ICT-related industries.** In the mergers and innovation process, enterprises in the ICT industry have a natural demand for digital asset acquisition, which may cause biased estimation results due to potential endogeneity. To alleviate this impact, we exclude ICT-related industries (computer, communication, and electronic equipment manufacturing industries) from the sample and rerun the regression analysis. The results are shown in Table 5, paragraph (3), and indicate that, under the control of endogeneity issues, the significant positive impact of corporate mergers and acquisitions on technological innovation is still reliable.

**Table 5. Robustness test of the impact of corporate mergers and acquisitions on technological innovation.**

| Variable | (1) | (2) | (3) | (4) |
|---|---|---|---|---|
| | Change regression method (Tobit) | Replace the dependent variable | Excluding ICT-related industries | Add industry control variables |
| | inno | innoa | inno | inno |
| Treat*Post | 0.228*** | 0.315*** | 0.397*** | 0.421*** |
| | (0.0257) | (0.0210) | (0.0261) | (0.0243) |
| Controls | YES | YES | YES | YES |
| Year FE | YES | YES | YES | YES |
| Firm FE | — | YES | YES | YES |
| Ind FE | YES | YES | YES | YES |
| Pro FE | YES | YES | YES | YES |
| Constant | -9.436*** | 0.0844*** | 0.0716*** | 1.143*** |
| | (0.272) | (0.0122) | (0.0177) | (0.207) |
| Observations | 68,698 | 68,698 | 56,938 | 68,698 |
| Pseudo $R^2$ | 0.373 | | | |
| R−squared | | 0.689 | 0.745 | 0.755 |

**5.3.4 Increasing the testing of industry control variables.** Wang et al. (2023) [47] noted that changes in industry demand may also cause changes in factor inputs, thereby affecting enterprise innovation. Therefore, we add variables that control for industry changes to the benchmark regression for further estimation, including the number of enterprises in the industry (*Firmnum*), the average company size in the industry (*Sizeind*), and the degree of industry competition (*HHI*). Represented by the industry Herfindahl index, the estimated results are still significantly positive, and the research conclusions remain robust. See Appendix 1 in S1 File for the meanings of the abbreviations.

**5.3.5 Parallel trend test.** We draw on the approach of Beck et al. (2010) [53] and Li et al. (2016) [49] to study multiple period DID events. The relative year information before and after mergers and acquisitions is included in the regression, and the annual changes in the technological innovation performance of enterprises before and after mergers and acquisitions are estimated. Based on this, the dynamic DID model is extended to:

$$Inno_{it} = \alpha_0 + \sum_{p=-5}^{6} \gamma_p did_{it+p} + \sum_{c=1}^{7} \alpha_c Control_{it+p} + Year_{FE} + Firm_{FE} + Ind_{FE} + Pro_{FE} + \varepsilon_{it} \quad (9)$$

Here, $did_{it+p} = Treat_{it+p}*Post_{it}$, and the year of a merger or acquisition is set as the base period (*current*), with 1 for the current period and 0 for other years. $Treat_{it+p}$ represents the relative $p$ year of a merger or acquisition for enterprise $i$ in year $t$, with a value of 1 for that year and 0 for other years. The meanings of the other variables are consistent with those in Eq (8). See Appendix 1 in S1 File for the meanings of the abbreviations.

Fig 1, which shows the regression results of Eq (9), indicates that the estimated coefficients $\gamma_p$ of the $did_{it+p}$ variable before the merger and acquisition ($p \leq -1$) are not significant. Therefore, there is no significant difference in the level of technological innovation between the

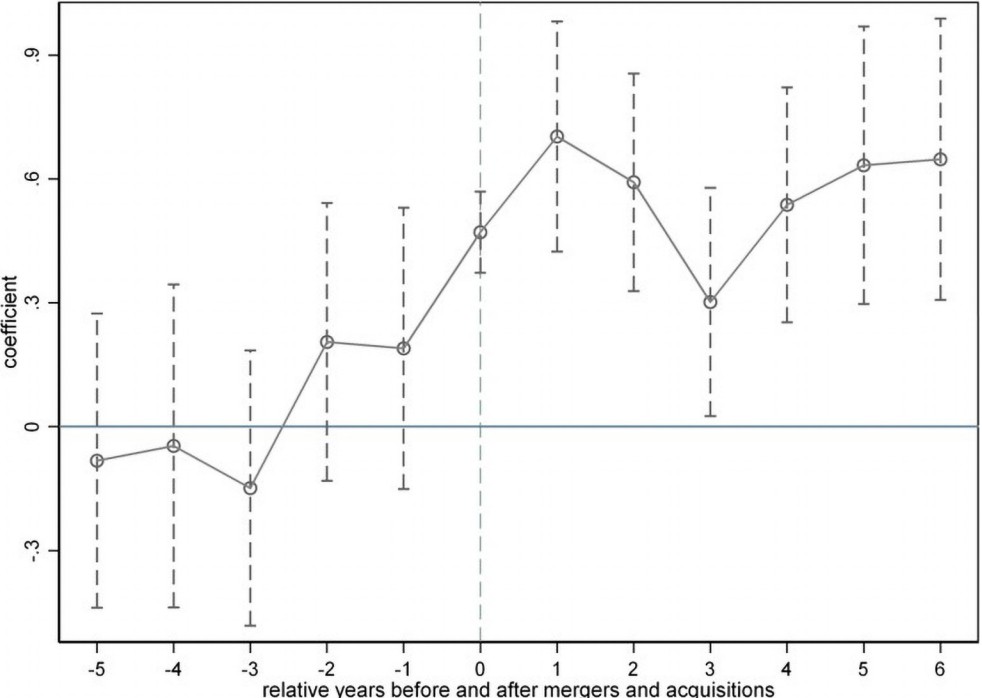

**Fig 1. Parallel trend test.**

control and treatment group samples before the merger or acquisition, satisfying the parallel trend test. The estimated coefficients for the years in which a merger occurred and subsequent years ($p = 0,1,...6+$)are significantly positive at least at the 5% level, indicating a relatively stable policy effect. The reason for the slight decrease in the regression coefficient two to three years after a merger or acquisition is that the company needs a period for technological research and innovation after completing the merger or acquisition in the same year. Alternatively, a processing time of two to three years occurs between the occurrence and completion of the merger. Fig 1 shows that the regression coefficient steadily increases in the fourth year after a merger or acquisition, indicating that mergers and acquisitions have a significant promoting effect on technological innovation.

**5.3.6 Placebo test.** We use the counterfactual method to design a placebo test for the impact of mergers and acquisitions on corporate technological innovation. The wrong time variable is set, and all merger and acquisition events are advanced by four years. The dummy variables *Treat_1*, *Treat_2*, *Treat_3*, and *Treat_4* are used to represent the year, two years, three years, and four years before the actual merger and acquisition of the enterprise, respectively. We observe whether mergers and acquisitions can have an impact on an enterprise's technological innovation during these four years [54]. If the dummy variable for new corporate mergers and acquisitions is still significant, then improvements in corporate technological innovation capability are likely caused by other factors. In contrast, if the results for technological innovation in the four years before mergers and acquisitions are not significant, then mergers and acquisitions are a key factor affecting corporate technological innovation. Columns (1) to (4) in Table 6 show that the estimated results for advancing mergers and acquisitions by four years are not significant, which verifies the robustness of the previous results. See Appendix 1 in S1 File for the meanings of the abbreviations.

We conduct a placebo test by randomly assigning merger and acquisition companies and merger and acquisition years and use $DID_{it}^{pseudo}$ instead of the core explanatory variable in Eq (8) for the empirical testing. The experiment is repeated 1000 times, and the test results are shown in Fig 2. Fig 2 shows that the estimated coefficients of the 1000 sampling results are all

**Table 6. Placebo test.**

| Variable | (1) | (2) | (3) | (4) |
|---|---|---|---|---|
| | inno | inno | inno | inno |
| did_1 | 0.0524 (0.0344) | | | |
| did_2 | | 0.0474 (0.0362) | | |
| did_3 | | | 0.0274 (0.0389) | |
| did_4 | | | | 0.000709 (0.0430) |
| Controls | YES | YES | YES | YES |
| Year FE | YES | YES | YES | YES |
| Firm FE | YES | YES | YES | YES |
| Ind FE | YES | YES | YES | YES |
| Pro FE | YES | YES | YES | YES |
| Constant | -5.646*** (0.615) | -5.682*** (0.613) | -5.721*** (0.613) | -5.743*** (0.614) |
| Observations | 68,698 | 68,698 | 68,698 | 68,698 |
| R−squared | 0.774 | 0.774 | 0.774 | 0.774 |

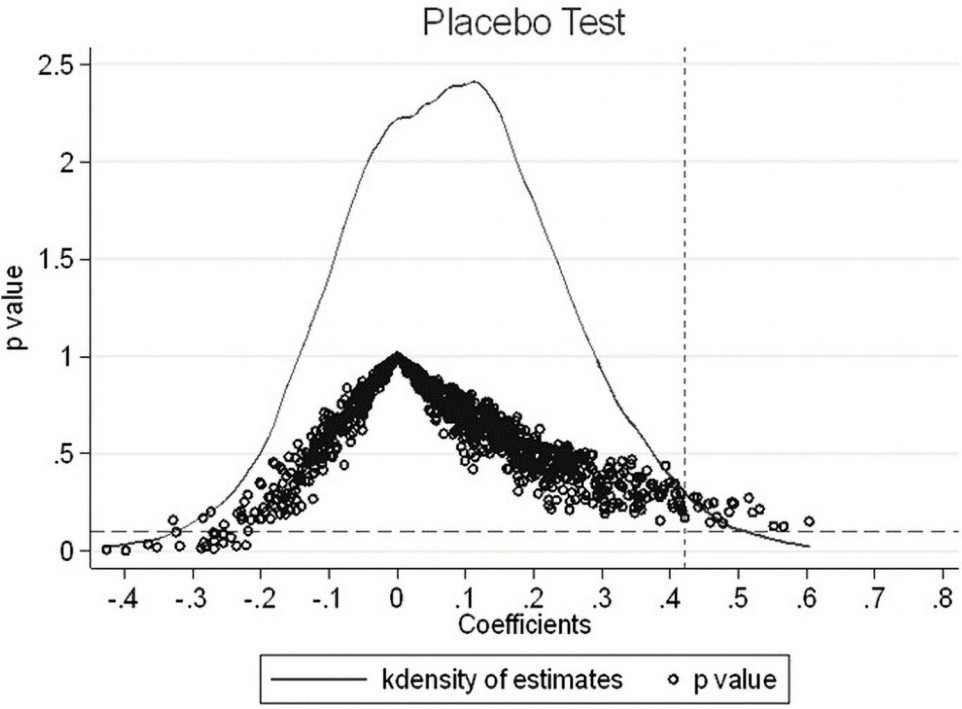

**Fig 2. Placebo test.**

approximately 0, and the *p* value distribution is generally above 0.1. Moreover, the true estimates in this article (column (4) of Table 3) are clearly outliers in the graph. The above placebo test results show that mergers and acquisitions are the main cause of technological innovation changes in enterprises.

**5.3.7 Endogeneity testing.**   Controlling for the fixed effects of time, region, industry, and enterprise can solve certain endogeneity problems. To further address endogeneity issues, we need to identify variables that are strongly correlated with corporate merger and acquisition behavior but are not related to corporate technological innovation.

Referring to the approach of Aiello et al. (2008) [55] and Zhou et al. (2012) [56], we quantify the likelihood of a merger occurring for the M&A firms in an M&A, construct a probit model to estimate the probability of a merger occurring for the M&A firms, and then estimate the fitted value of its estimate as an instrumental variable for model (8) [57].

$$Prob\{treat = 1\} = \delta_0 + \delta_1 lnnum_i + \sum_{c=1}^{7} \alpha_c Control_{it} + Year_{FE} + Firm_{FE} + Ind_{FE} + Pro_{FE} + \varepsilon_{it} \quad (10)$$

Here, $lnnum_i$ represents the logarithm of 1 plus the annual cumulative value of relevant articles, conferences, newspapers, etc., identified by the China National Knowledge Infrastructure (CNKI) with "mergers and acquisitions" as the theme keyword. See Appendix 1 in S1 File for the meanings of the abbreviations. The number of studies on mergers and acquisitions can significantly affect the likelihood of a company engaging in mergers and acquisitions. When a greater number of studies include mergers and acquisitions as the theme, the mergers and acquisitions activity is greater but does not directly affect the company's level of technological innovation ability. Therefore, choosing this variable as an important influencing factor for a company's merger and acquisition behavior is reasonable. Table 7 shows that the validation

**Table 7. Endogeneity test results.**

| Variable | (1) | (2) | (3) | (4) |
|---|---|---|---|---|
| | Panel Tool Variables | | 2SLS | |
| | inno | inno | inno | inno |
| Treat*Post | 0.539*** | 0.435*** | 0.503*** | 0.495*** |
| | (0.0118) | (0.0235) | (0.0765) | (0.0771) |
| Controls | YES | YES | YES | YES |
| Year FE | NO | YES | NO | YES |
| Firm FE | NO | YES | NO | YES |
| Ind FE | NO | YES | NO | YES |
| Pro FE | NO | YES | NO | YES |
| F Value | 5235.10 | 390.54 | 726.50 | 379.57 |
| Kleibergen−Paap rkLM statistic | — | — | 1087.724 [0.0000] | 1210.106 [0.0000] |
| Kleibergen−Paap rkWald F statistic | — | — | 726.504 {19.93} | 1489.946 {16.38} |
| Hansen J statistic | — | — | 9.1950 | 0.0000 |
| Observations | 68,698 | 68,698 | 68,698 | 68,698 |
| R−squared | 0.433 | 0.752 | 0.430 | 0.460 |

Note: () is the standard deviation of the coefficient, [] is the corresponding statistical P value, {} is the critical value of a 10% deviation in Stock−Yogo, and the *Hansen J statistic* = 0 indicates that the identification is accurate; that is, the number of instrumental variables equals the number of endogenous variables.

results obtained using the instrumental variable method indicate that the estimated coefficient of mergers and acquisitions on the level of technological innovation is significant at the 1% level. After introducing the instrumental variables, the estimation results once again verify the positive incentive effect of mergers and acquisitions on technological innovation improvements based on the rejection of under identification, weak instrumental variables, and over-identification, indicating that the original conclusion is relatively robust. Therefore, the results of this article are not affected by the endogeneity of corporate mergers and acquisitions.

## 5.4 Impact mechanism testing

To further examine the impact mechanism of mergers and acquisitions on corporate technological innovation, we first replace the original dependent variable, technological innovation output, with a new dependent variable and regress it on the original core explanatory variable [58, 59]. The specific regression equation is as follows:

$$m_{it}^{\lambda} = \alpha^{\lambda} Treat_{it}*Post_{it} + \sum_{c=1}^{7} \alpha_c Control_{it} + Year_{FE} + Firm_{FE} + Ind_{FE} + Pro_{FE} + \varepsilon_{it}^{\lambda} \quad (11)$$

Here, $m_{it}^{\lambda}(\lambda = 1, 2, 3, 4)$ represents the measurement indicators of the four types of impact mechanism variables selected by enterprise $i$ in year $t$, and the meanings of the other variables are the same as in Eq (8). Then, all of the influencing mechanism variables are added as new explanatory variables to the benchmark regression to test whether the different mechanism variables can improve the level of enterprise technological innovation. See Appendix 1 in S1 File for the meanings of the abbreviations. The specific regression equation is as follows:

$$inno_{it} = \alpha_0' + \alpha_1' Treat_{it}*Post_{it} + \sum_{\lambda=1}^{4} \gamma^{\lambda} m_{it}^{\lambda} + \sum_{c=1}^{7} \alpha_c Control_{it} + Year_{FE} + Firm_{FE} + Ind_{FE} + Pro_{FE} + \varepsilon_{it} \quad (12)$$

**Table 8. Impact mechanism test results.**

| Variable | (1) | (2) | (3) | (4) | (5) |
|---|---|---|---|---|---|
| | TFP_LP | Digit | ias | lerner | inno |
| Treat*Post | 1.442*** | 0.473*** | 0.295*** | 0.0121*** | 0.306*** |
| | (0.0561) | (0.0239) | (0.0484) | (0.00371) | (0.0250) |
| TFP_LP | | | | | 0.0324*** |
| | | | | | (0.00247) |
| Digit | | | | | 0.130*** |
| | | | | | (0.00935) |
| ias | | | | | 0.0206*** |
| | | | | | (0.00454) |
| lerner | | | | | 0.123* |
| | | | | | (0.0720) |
| Controls | YES | YES | YES | YES | YES |
| Year FE | YES | YES | YES | YES | YES |
| Firm FE | YES | YES | YES | YES | YES |
| Ind FE | YES | YES | YES | YES | YES |
| Pro FE | YES | YES | YES | YES | YES |
| Constant | 1.083*** | 0.0250 | 0.00184 | -0.0299*** | 0.0279* |
| | (0.0294) | (0.0153) | (0.0271) | (0.00369) | (0.0160) |
| Observations | 68,698 | 68,698 | 68,698 | 68,698 | 68,698 |
| R−Squared | 0.836 | 0.833 | 0.978 | 0.688 | 0.761 |

**5.4.1 Production efficiency effect.** This section verifies whether mergers and acquisitions promote enterprises' level of technological innovation by improving their production efficiency. We draw on the approach of Wang (2023) [60] and calculate total factor productivity to represent the production efficiency of enterprises for mechanism testing. The regression results are shown in column (1) of Table 8. The estimated coefficient of the core explanatory variable $Treat_{it}*Post_{it}$ is 1.442 and significant, indicating that mergers and acquisitions can improve enterprise total factor productivity. Moreover, based on the estimation results in column (5) of Table 8, the estimated coefficient of total factor productivity is 0.0324, which is significantly positive at the 1% level, indicating that improving total factor productivity can promote technological innovation in enterprises. Therefore, hypothesis 2 holds.

**5.4.2 Digital knowledge effect.** In this section, we verify whether mergers and acquisitions improve an enterprise's technological innovation level through an increase in its digital knowledge base. We refer to the approach of Wu et al. (2021) [61] and Lim et al. (2020) [62] and use digital transformation and intangible assets to represent enterprises' basic digital knowledge levels. The data are obtained from the annual reports of various listed companies and the CSMAR database, and the types of data are denoted as Digit and ias. The following digital transformation indicators are obtained. First, the annual reports of Chinese A-share listed companies are collected and organized by a Python crawler, and all content was extracted from the data pool used for feature word screening. Second, classical literature, policy documents and research reports with the theme of digital features are referred to, and specific keywords about digital transformation are summarized, mainly including cloud computing technology, big data technology, artificial intelligence technology, blockchain technology and feature words related to the use of digital technology in practice. Third, the word frequency of the feature words is counted to obtain the total word frequency, and the natural logarithm is taken by adding 1 to the total word frequency as a measure of digital transformation; intangible assets are measured by adding 1 to the natural logarithm of the ratio of intangible assets to total assets. The estimated results in columns (2), (3), and (5) of Table 8 lead to the

conclusion that corporate mergers and acquisitions can enrich the digital knowledge stock of enterprises by enhancing their digital transformation and intangible assets, thereby positively affecting improvements in technological innovation. Therefore, hypothesis 3 proposed in this article is validated.

**5.4.3 Market power effect.** We refer to the approach of Peress (2010) [63] and Datta et al. (2011) [64] and use the Lerner index (lerner) to measure a company's market power, calculated as (operating revenues—costs)/operating revenues. The regression results in columns (4) and (5) of Table 8 indicate that mergers and acquisitions can significantly improve a company's Lerner index, which also significantly increases the number of patent applications. Mergers and acquisitions are observed to promote technological innovation by enhancing a company's market power. This conclusion is consistent with the findings of Ringel et al. (2017) [65]. Therefore, hypothesis 4 holds.

## 5.5 Heterogeneity analysis

At this point, we have tested and analyzed how corporate mergers and acquisitions can significantly enhance technological innovation and how to enhance technological innovation. However, are the results of mergers and acquisitions influenced by factors such as the type of merger and acquisition and the nature of the acquiring company? Therefore, in this section, we analyze the impact of mergers and acquisitions using different corporate characteristics on technological innovation from two perspectives: the heterogeneity of enterprise transactions and the heterogeneity of enterprise ownership.

**5.5.1 Heterogeneity of enterprise transactions.** We mainly explore transaction heterogeneity from two perspectives: whether cross-border mergers and acquisitions occur and the number of merger and acquisition transactions. First, merger and acquisition events are divided into domestic and cross-border mergers and acquisitions for sample regression. The regression results are shown in columns (1) and (2) of Table 9. The results indicate that both domestic and cross-border mergers and acquisitions can promote enterprise technological innovation to a certain extent. This discovery is highly in line with economic intuition. The

**Table 9. Heterogeneity analysis.**

| Variable | (1) | (2) | (3) | (4) | (5) | (6) |
|---|---|---|---|---|---|---|
| | Transaction heterogeneity | | | | Enterprise heterogeneity | |
| | Domestic M&A | Cross-border M&A | Large amount | Small amount | State-owned enterprise | Non-state-owned enterprises |
| | inno | inno | inno | inno | inno | inno |
| Treat*Post | 0.277*** | 0.263* | 0.274*** | 0.330*** | 0.0599 | 0.452*** |
| | (9.67) | (1.69) | (0.0481) | (0.0496) | (0.0496) | (0.0284) |
| Controls | YES | YES | YES | YES | YES | YES |
| Year FE | YES | YES | YES | YES | YES | YES |
| Firm FE | YES | YES | YES | YES | YES | YES |
| Ind FE | YES | YES | YES | YES | YES | YES |
| Pro FE | YES | YES | YES | YES | YES | YES |
| Constant | 0.254*** | 0.552*** | 0.204*** | 0.169*** | -4.733*** | 0.0124 |
| | (6.93) | (3.60) | (0.0699) | (0.0478) | (1.544) | (0.0120) |
| Observations | 24,570 | 910 | 10,402 | 7,168 | 14,042 | 54,515 |
| R–squared | 0.742 | 0.804 | 0.748 | 0.726 | 0.800 | 0.745 |
| Intergroup coefficient test for Treat*Post [P value] | -0.014 [0.476] | | 0.056** [0.046] | | 0.393*** [0.000] | |

acquiring and target companies involved in domestic mergers and acquisitions have similar cultural, business environment, and business models. Therefore, mergers and acquisitions can be integrated at lower costs, allowing synergies to be realized faster and technological innovation to be promoted. Cross-border mergers and acquisitions can provide more diverse merger and acquisition resources for Chinese companies to absorb many high-quality foreign companies, thereby generating greater synergies and technology spillover effects.

Second, in this article, we use the median transaction amount as the standard for dividing enterprises into large transaction mergers and acquisitions and small transaction mergers and acquisitions and regressing them separately. The regression results in columns (3) and (4) of Table 9 show that both large and small transaction mergers and acquisitions can promote enterprise technological innovation. Moreover, the intergroup coefficient difference test results for the core explanatory variable *Treat*Post* also indicate that transaction heterogeneity is significant. This result is attributed to the fact that acquirers, whether they acquire large firms or small firms, are able to acquire new knowledge from the acquired firms and rationalize the use of this knowledge resource for technological innovation.

**5.5.2 Heterogeneity of enterprise ownership.** We mainly examine the impact of corporate mergers and acquisitions on technological innovation from the perspective of the nature of ownership of M&A firms. The regression results are shown in columns (5) and (6) of Table 9. The results indicate that when an M&A enterprise is state-owned, its merger and acquisition behavior does not significantly promote its innovation output. However, when an M&A enterprise is non-state-owned, mergers and acquisitions can significantly promote its technological innovation. Therefore, we believe that the reason for this result may be that state-owned enterprises have relatively cumbersome organizational structures and overall low efficiency, resulting in a slow enterprise technological innovation process. In contrast, non-state-owned enterprises are self-reliant on profits and losses and are more eager to gain market opportunities and advantages by accelerating technological innovation to maximize corporate interests.

## 6. Discussion

The current research results are quite rich, and this paper builds on this foundation and further enriches the body of knowledge by focusing on Chinese listed companies.

First, at present, academics have conducted many theoretical and empirical studies on the relationship between mergers and acquisitions and technological innovation, but the conclusions are significantly divergent. Taking Chinese A-share listed companies as the research object, this paper finds that corporate mergers and acquisitions have a positive effect on technological innovation, which still holds after a parallel trend test and a placebo test, and further verifies that corporate mergers and acquisitions are the main driving force of technological innovation, which supports the conclusions of the studies by Wu et al. (2021) [25] and Chen et al. (2024) [66]. Additionally, this paper finds that the enhancement effect of mergers and acquisitions on technological innovation persists in the long term and becomes increasingly obvious as the postmerger integration process advances. As shown in Fig 2, $\gamma_6 = 0.6477$ is significant at the 1% level, indicating that six years after a merger and acquisition occurs, it can still significantly enhance the technological innovation of the firm, and the economic significance is high.

Second, current scholars focusing on high-tech enterprises and family-owned enterprises believe that mergers and acquisitions can promote technological innovation, which is mainly realized through saving transaction costs, improving enterprise productivity [66], reducing R&D risks, and improving R&D efficiency (Ye et al., 2024) [67]. This paper verifies that

improving productivity is an important path, which supports the current research. Additionally, this paper finds that mergers and acquisitions can also promote technological innovation through two paths, specifically, enriching the level of digital knowledge and enhancing market power, of which the enhanced effect of enterprise digital transformation has the strongest explanatory power, suggesting that it is an important pathway for enhancing technological innovation. This further reveals the mechanism by which mergers and acquisitions promote technological innovation.

Third, current research suggests that firms with high productivity [8], strong resource reorganization capabilities [54], and few financing constraints [50] are more suitable for mergers and acquisitions. This study contributes by showing that when a merger and acquisition firm is a non-state-owned enterprise, its mergers and acquisitions have a significant positive effect on technological innovation, and the mergers and acquisitions of state-owned enterprises do not have a significant effect on technological innovation. Domestic mergers and acquisitions play a more pronounced role in promoting technological innovation than cross-border mergers and acquisitions, and small-dollar mergers and acquisitions play a more pronounced role in promoting technological innovation than large-dollar mergers and acquisitions. This study further reveals the characteristics of enterprises that are more suitable for promoting technological innovation through mergers and acquisitions from the perspectives of enterprise ownership, merger and acquisition type and merger and acquisition intensity.

Nonetheless, this study has some limitations. First, this study focuses on data from listed companies in China and attempts to consider nonlisted companies in the subsequent period. Second, this study observes the acquiring firms in merger and acquisition transactions, and the traits of the acquired firms, as with the other participants in merger and acquisition transactions, may also have an impact on technological innovation. Third, the positive impact of M&A on technological innovation in enterprises has been demonstrated in many cases around the world. For example, Qualcomm in the United States successfully acquired Nuvia, a chip architecture design company, and utilized Nuvia's advanced design concepts and technological advantages to build a chip that surpassed the performance of Apple's A17Pro and Snapdragon 8gen3 in 2023; this greatly enhanced the technological innovation level of the enterprise. As an innovation-driven, research and development-based healthcare multinational, Roche Switzerland broadened its molecular diagnostics portfolio through the acquisitions of state-of-the-art U.S.-based bioengineering companies Genentech and GeneMark Diagnostics as a means of driving technological innovation and product development. Through the acquisition of Viv Labs and Harman International Industries, South Korea's Samsung has achieved technological breakthroughs in intelligent voice interaction and autonomous driving, completing the industry chain from software to hardware. However, there are significant differences in M&A behaviors and external conditions in different countries, specifically in terms of economic level, cultural variability, and political influence. Therefore, the impact of M&A on the technological innovation of enterprises is bound to vary among countries. Accordingly, future research can be carried out in the following three ways. First, nonlisted companies can be used as the research object in the subsequent period to comprehensively understand the impact of mergers and acquisitions on technological innovation in Chinese enterprises. Second, the acquired enterprises can be used as the research object to further enrich the body of knowledge on the impact of mergers and acquisitions on technological innovation. Third, in the future, it will be necessary to consider enterprises from countries other than China as research objects and to further compare and analyze the different mechanisms and paths of the influence of M&A on technological innovation among different countries.

## 7. Conclusions and recommendations

### 7.1 Conclusions

This paper constructs a mathematical model, uses panel data on China's A-share listed companies in Shanghai and Shenzhen from 2007 to 2020, constructs a theoretical framework for mergers and acquisitions to promote technological innovation, utilizes the Difference-in-Differences method to empirically test the impact of mergers and acquisitions on technological innovation, and further examines the path of the role of the two and the impact of heterogeneity. The research presented in this paper finds the following.

First, corporate mergers and acquisitions can significantly promote technological innovation in enterprises. After we change the regression method, explain the dependent variable, remove ICT-related industries, and add industry control variables for robustness testing, this conclusion still holds. The analysis of the dynamic effects shows that corporate mergers and acquisitions have long-term and sustained effects on enhancing technological innovation. The regression coefficient slightly decreases two to three years after a merger because of the company's need for technology absorption, research and development, and innovation after completing a merger in the same year or the existence of a two- to three-year processing time between the occurrence and completion of a merger. In the fourth year after a merger and acquisition transaction, the promoting effect on technological innovation steadily increased annually, indicating that corporate mergers and acquisitions have a significant and long-term promoting effect on technological innovation.

Second, the production efficiency effect, knowledge effect, and market power effect are the three important mechanisms through which enterprise mergers and acquisitions enhance technological innovation. Improvements in production efficiency indicate a decrease in enterprises' marginal costs, and such improvements generated by mergers and acquisitions mainly occupy a market advantage position from the cost side. Given the gradual integration of intangible assets, such as digital technology knowledge, management experience, and marketing networks, the knowledge stock of merger and acquisition enterprises increases, which helps improve their resource allocation efficiency and generate synergistic effects. Enterprises reduce the number of market competitors through mergers and acquisitions, improve their market competitiveness, and expand their market power. Further empirical results indicate that mergers and acquisitions can promote technological innovation by improving enterprises' total factor productivity, enriching their digital knowledge, and enhancing their market power.

Third, the impact of mergers and acquisitions on technological innovation exhibits heterogeneity due to the nature of corporate ownership. Compared to state-owned enterprises, non-state-owned enterprises have a more significant effect on achieving technological innovation through mergers and acquisitions. Regarding merger and acquisition transaction types, whether domestic or cross-border, large- or small-scale mergers and acquisitions can enhance a company's technological innovation. When mergers and acquisitions meet the requirements of domestic mergers and acquisitions and small transaction amounts and when the main merging enterprise is a non-state-owned enterprise, the technological innovation enhancement effect of enterprise mergers and acquisitions becomes more significant.

### 7.2 Policy recommendations

Based on the above findings, the following recommendations can be made.

First, the government should provide a strong policy environment and financing support for mergers and acquisitions to provide a source of power for enterprise technological innovation. Banks and other financial institutions should be guided to broaden the financing

channels for mergers and acquisitions and provide governmental financing guarantees for enterprises to carry out mergers and acquisitions. Additionally, appropriate policy subsidies and tax exemptions and reductions for mergers and acquisitions should be provided to minimize the risks and costs faced by enterprises in mergers and acquisitions. Enterprises should make full use of the government's supportive policies to support mergers and acquisitions, focus on R&D innovation after mergers and acquisitions, increase R&D investment, cultivate professional and technical talent, and truly transform the technological resources acquired through mergers and acquisitions into the enterprise's own technological strength. For example, in 2014, the State Council of China issued the Opinions on Further Optimizing the Market Environment for Corporate Mergers and Restructuring, which explicitly encourages financial institutions to carry out mergers and restructuring of financing for businesses; additionally, various types of financial investment entities can participate in mergers and restructuring by setting up equity investment funds, venture capital funds, industrial investment funds, merger and acquisition funds, etc. The use of diversified fundraising channels can help reduce the financial constraints on merging and acquiring enterprises, safeguard further R&D investment by enterprises, and thus promote their innovative activities. According to CSMAR database statistics, in 2014, the number of M&A cases announced by A-share listed Chinese enterprises in Shanghai and Shenzhen was 5,689, and the number of patent applications by enterprises was 16,955. After the opinion was put forward, the number of M&A cases reached 7,945 in 2015, and the number of patent applications was 26,766, showing growth rates of 40% and 58%, respectively.

Second, the government should build a digital resource platform; promote the convergence of innovative resources; accelerate the efficient circulation of data, knowledge and other factors of production; strengthen the joint cultivation of schools and enterprises; expand the channels for attracting talent from home and abroad; increase the introduction and training of technical talent; open up the chain of talent from enterprises, universities and scientific research institutes; establish industry-academia-research cooperation teams; gather resources from industry-academia-research; systematically promote the technological research of key areas; give enterprises priority access to new products; and utilize their own influence and social credibility to promote quality products. Enterprises should absorb talent around the core technology after the completion of mergers and acquisitions, promote the process of technology absorption and reinnovation, accelerate the transformation of core technology, emphasize the development of new products, accelerate the promotion of new products, create unique brands belonging to the enterprise, and enhance the competitiveness of the market. For example, Shanxi Province in China, with its rich resources of universities and research institutes, has been steadily moving forward in recent years in terms of the intensity of R&D investment in general and by universities, the number of geographic contracts for technology output in the technology market, the amount of cooperation between enterprises and universities and research institutes, and the number of papers jointly published by enterprises and universities and research institutes. These efforts have laid a solid foundation of science and technology and industrial innovation for the cultivation and development of high-quality products. Regarding enterprises, Tencent successively acquired Riot Games, Epic Games and Supercell and then quickly absorbed resources and launched game products such as League of Legends, Clash of Clans, and Clash Royale; it thereby smoothly entered the game industry and seized the market position, and it has now become the world's largest game publisher.

Third, the government should increase the promotion of technological mergers and acquisitions by enterprises, focus on the selection of targets for mergers and acquisitions by enterprises, and encourage enterprises to achieve technological innovation through technological mergers and acquisitions. It should implement merger and acquisition encouragement policies

in a prudent manner, avoid concentrating too many merger and acquisition resources on large enterprises and state-owned enterprises, and provide a level playing field for the industry; in particular, it should guide small and non-state-owned enterprises to cultivate their own core competencies, which are symbolized by their technologies and patents. Enterprises should select M&A objects by strategically considering technological innovation and the actual situation. In particular, enterprises with technology as the main driver of profit should give full play to their absolute advantages and should pursue merges and acquisitions of other technology enterprises to combine strengths; after completing a merger or acquisition, the company should immediately begin the integration and utilization of innovative resources, accelerate the process of technological innovation, and ultimately achieve innovation in key technological fields. For example, Huawei Technologies Co., Ltd. acquired Caliopa, a silicon photonics technology developer in the data communication and telecom markets, and then quickly absorbed the core technology and innovation resources of the target enterprise and leveraged new-generation information technologies such as big data, the Internet of Things, and blockchain to enhance its independent innovation capability, make breakthroughs in core technologies, and solve bottleneck problems. Finally, Huawei successfully researched and developed its own 5G base station core chip in September 2023, released the Kirin 9000S processor, achieved the localization of the 5G chip, and thereby occupied a leading position in global 5G technology.

## Supporting information

**S1 File. Appendix 1.**
(ZIP)

**S2 File. Original data.**
(XLSX)

## Author Contributions

**Conceptualization:** Yujiao Bai, Hao Zhang.

**Formal analysis:** Yujiao Bai, Hao Zhang.

**Methodology:** Yujiao Bai, Hao Zhang.

**Software:** Yujiao Bai.

**Writing – original draft:** Yujiao Bai.

**Writing – review & editing:** Yujiao Bai, Hao Zhang.

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
