## [Decision Letter · Decision Letter 0]

15 May 2024

PONE-D-24-14261Research on the mechanism and path of digital transformation driving technological innovation: Based on the perspective of enterprise mergers and acquisitionsPLOS ONE

Dear Dr. ZHANG,

Thank you for submitting your manuscript to PLOS ONE. After careful consideration, we feel that it has merit but does not fully meet PLOS ONE’s publication criteria as it currently stands. Therefore, we invite you to submit a revised version of the manuscript that addresses the points raised during the review process.

We look forward to receiving your revised manuscript.

Kind regards,

Marcelo Dionisio

Academic Editor

PLOS ONE

Journal Requirements:

"This research was supported by Anhui Province Excellent Youth Research Project in Universities, grant number 2023AH030082."

3. We note that your Data Availability Statement is currently as follows: 

"All relevant data are within the manuscript and its Supporting Information files."

6. Please upload a copy of Supporting Information (S1 File. S1 Appendix.) which you refer to in your text on page 55 (in PDF format). 

7. We notice that your supplementary figures are uploaded with the file type 'Figure'. Please amend the file type to 'Supporting Information'. Please ensure that each Supporting Information file has a legend listed in the manuscript after the references list.

**Additional Editor Comments:**

Based on the comments of both reviewers and my own assessment, I am recommending a major revision decision. I urge you to review your introduction and abstract, and make it more structured, offering a clear perspective of your problem, gap, research question, method and contribution, all in a connected an dlogical order. I also highlight the necessity to translate the values in Tuan to dollar, to make it a more clear reference. Please, carefully assess the comments from reviewer # 1, and if possible, perfomr a professional English review.

Reviewers' comments:

Reviewer's Responses to Questions

**Comments to the Author**

1. Is the manuscript technically sound, and do the data support the conclusions?

Reviewer #1: Yes

Reviewer #2: Partly

2. Has the statistical analysis been performed appropriately and rigorously? 

Reviewer #1: Yes

Reviewer #2: Yes

3. Have the authors made all data underlying the findings in their manuscript fully available?

Reviewer #1: Yes

Reviewer #2: No

4. Is the manuscript presented in an intelligible fashion and written in standard English?

Reviewer #1: No

Reviewer #2: Yes

5. Review Comments to the Author

Reviewer #1: 1. The Abstract is too long. Need to shorten the abstract.

2. Rearrange abstract: Include problem statements, objectives, methods, findings, and recommendations step by step.

3. The very first paragraph of introduction section is not clear as mentioned ;the 2023 work report of Government of state council further highlights what? and how will you relate it with your current study. i think it is a poor start of introduction chapter.

4. Re arrange the structure of Introduction section

5. Provide a separate section of limitations of current work.

6. Add seperate table for abbreviations used in this research work.

7. English used in this work is too weak, found grammatically error which has greater impact on this current paper and also can degrade the value of this journal.

8. References are not in alphabetical order. Rearrange them.

Reviewer #2: The research content of this study is rich. However, the lack of clarity of the research question is the major flaw of the study.Therefore, MAJOR revision has to be done before this manuscript could be accepted for publication in the PLOS ONE.

6. PLOS authors have the option to publish the peer review history of their article (what does this mean?). If published, this will include your full peer review and any attached files.

Reviewer #1: No

Reviewer #2: No

---

## [Author Response · Author response to Decision Letter 0]

19 Jun 2024

Dear Reviewers

Thank you for your letter and constructive comments on our manuscript, "Research on Mechanisms and Paths of Digital Transformation-Driven Technological Innovation: A Perspective Based on Corporate Mergers and Acquisitions" (PONE-D-24-14261). All of us authors have carefully read the comments you have given us and discussed and revised them one by one. The list of revisions is shown in the "Response to Reviewers". In addition, we have resubmitted a new manuscript in its revised state with the changes highlighted in red. Please feel free to let us know if there are any wrong answers or questions in the manuscript.

---

## [Decision Letter · Decision Letter 1]

18 Jul 2024

PONE-D-24-14261R1Research on the impact of enterprise mergers and acquisitions on technological innovation: An empirical analysis based on Chinese listed enterprisesPLOS ONE

Dear Dr. ZHANG,

Thank you for submitting your manuscript to PLOS ONE. After careful consideration, we feel that it has merit but does not fully meet PLOS ONE’s publication criteria as it currently stands. Therefore, we invite you to submit a revised version of the manuscript that addresses the points raised during the review process.

We look forward to receiving your revised manuscript.

Kind regards,

Marcelo Dionisio

Academic Editor

PLOS ONE

Journal Requirements:

Additional Editor Comments:

I kindly suggest yoou follow reviewer's 3 comments and suggenstions in one ast round of revision.

Reviewers' comments:

Reviewer's Responses to Questions

**Comments to the Author**

1. If the authors have adequately addressed your comments raised in a previous round of review and you feel that this manuscript is now acceptable for publication, you may indicate that here to bypass the “Comments to the Author” section, enter your conflict of interest statement in the “Confidential to Editor” section, and submit your "Accept" recommendation.

Reviewer #3: All comments have been addressed

Reviewer #4: All comments have been addressed

2. Is the manuscript technically sound, and do the data support the conclusions?

Reviewer #3: Yes

Reviewer #4: Yes

3. Has the statistical analysis been performed appropriately and rigorously? 

Reviewer #3: (No Response)

Reviewer #4: Yes

4. Have the authors made all data underlying the findings in their manuscript fully available?

Reviewer #3: Yes

Reviewer #4: Yes

5. Is the manuscript presented in an intelligible fashion and written in standard English?

Reviewer #3: No

Reviewer #4: Yes

6. Review Comments to the Author

Reviewer #3: The manuscript presents a thorough analysis of the impact of mergers and acquisitions on technological innovation, supported by a robust empirical methodology and theoretical framework.

1. Suggest incorporating comparative analyses with companies from different countries to provide broader insights into the impact of mergers and acquisitions on technological innovation in diverse contexts.

2. Consider incorporating case studies or real-world examples to illustrate how successful implementation of these policy recommendations can lead to tangible outcomes in terms of technological innovation and business growth.

Reviewer #4: Dear Authors,

I think the paper is interesting, the empirical methodology and results are sound and overall the paper in it's current form is good. I have one comment concerning the last policy recommendation. Your paper does show gains from M&A and firm consolidations, however there are known welfare costs to mergers and market monopolization. So I would be careful in recommending a blanket easing of mergers. I suggest you reformulate this last recommendation, to reflect something along the lines that the policy makers should internalize the technological innovation aspect (documented in your paper) in decision making concerning mergers and acquisition.

7. PLOS authors have the option to publish the peer review history of their article (what does this mean?). If published, this will include your full peer review and any attached files.

Reviewer #3: No

Reviewer #4: No

---

## [Author Response · Author response to Decision Letter 1]

6 Aug 2024

Responses to all reviewer and editorial comments can be found in the “Response to Reviewers” word file.

---

## [Editor Report · Decision Letter 2]

15 Aug 2024

Research on the impact of enterprise mergers and acquisitions on technological innovation: An empirical analysis based on listed Chinese enterprises

PONE-D-24-14261R2

Dear Dr. ZHANG,

We’re pleased to inform you that your manuscript has been judged scientifically suitable for publication and will be formally accepted for publication once it meets all outstanding technical requirements.

Kind regards,

Pu-yan Nie

Academic Editor

PLOS ONE
---

## [Editor Report · Acceptance letter]

20 Aug 2024

PONE-D-24-14261R2 

PLOS ONE

Dear Dr. ZHANG, 

I'm pleased to inform you that your manuscript has been deemed suitable for publication in PLOS ONE. Congratulations! Your manuscript is now being handed over to our production team.

Kind regards, 

on behalf of

Dr. Pu-yan Nie 

Academic Editor

PLOS ONE